# The Convolution Exponential and Generalized Sylvester Flows

**Emiel Hoogeboom**
UvA-Bosch Delta Lab
University of Amsterdam
The Netherlands
e.hoogeboom@uva.nl

**Victor Garcia Satorras**
UvA-Bosch Delta Lab
University of Amsterdam
The Netherlands
v.garciasatorras@uva.nl

**Jakub M. Tomczak**
Vrije Universiteit Amsterdam
The Netherlands
j.m.tomczak@vu.nl

**Max Welling**
UvA-Bosch Delta Lab
University of Amsterdam
The Netherlands
m.welling@uva.nl

## Abstract

This paper introduces a new method to build linear flows, by taking the exponential of a linear transformation. This linear transformation does not need to be invertible itself, and the exponential has the following desirable properties: it is guaranteed to be invertible, its inverse is straightforward to compute and the log Jacobian determinant is equal to the trace of the linear transformation. An important insight is that the exponential can be computed implicitly, which allows the use of convolutional layers. Using this insight, we develop new invertible transformations named *convolution exponentials* and *graph convolution exponentials*, which retain the equivariance of their underlying transformations. In addition, we generalize Sylvester Flows and propose *Convolutional Sylvester Flows* which are based on the generalization and the convolution exponential as basis change. Empirically, we show that the convolution exponential outperforms other linear transformations in generative flows on CIFAR10 and the graph convolution exponential improves the performance of graph normalizing flows. In addition, we show that Convolutional Sylvester Flows improve performance over residual flows as a generative flow model measured in log-likelihood.

## 1 Introduction

Deep generative models aim to learn a distribution $p_X(\mathbf{x})$ for a high-dimensional variable $\mathbf{x}$. Flow-based generative models (Dinh et al., 2015, 2017) are particularly attractive because they admit exact likelihood optimization and straightforward sampling. Since normalizing flows are based on the change of variable formula, they require the flow transformation to be *invertible*. In addition, the Jacobian determinant needs to be tractable to compute the likelihood.

In practice, a flow is composed of multiple invertible layers. Since the Jacobian determinant is required to compute the likelihood, many flow layers are triangular maps, as the determinant is then the product of the diagonal elements. However, without other transformations, the composition of triangular maps will remain triangular. For that reason, triangular flows are typically interleaved with *linear flows* that mix the information over dimensions. Existing methods include permutations (Dinh et al., 2015) and $1 \times 1$ convolutions (Kingma and Dhariwal, 2018) but these do not operate over feature maps *spatially*. Alternatives are emerging convolutions (Hoogeboom et al., 2019a) and

Figure 1: Visualization of the equivalent matrix exponential $\exp(\mathbf{M})$ where $\mathbf{M}$ represents a 2d convolution on a $1 \times 5 \times 5$ input (channel first). In this example the computation is explicit, however in practice the exponential is computed implicit and the matrices $\mathbf{M}$ and $\exp(\mathbf{M})$ are never stored.

periodic convolutions (Finzi et al., 2019; Karami et al., 2019). However, periodicity is generally not a good inductive bias for images, and emerging convolutions are autoregressive and their inverse is solved iteratively over dimensions.

In this paper, we introduce a new method to construct invertible transformations, by taking the *exponential* of any linear transformation. The exponential is always invertible, and computing the inverse and Jacobian determinant is straightforward. Extending prior work Goliński et al. (2019), we observe that the exponential can be computed *implicitly*. As a result, we can take the exponential of linear operations for which the corresponding matrix multiplication would be intractable. The canonical example of such a transformation is a convolutional layer, using which we develop a new transformation named the *convolution exponential*. In addition we propose a new residual transformation named *Convolutional Sylvester Flow*, a combination of a generalized formulation for Sylvester Flows, and the convolution exponential as basis change. Code for our method can be found at: https://github.com/ehoogeboom/convolution_exponential_and_sylvester

## 2 Background

Consider a variable $\mathbf{x} \in \mathbb{R}^d$ and an invertible function $f : \mathbb{R}^d \to \mathbb{R}^d$ that maps each $\mathbf{x}$ to a unique output $\mathbf{z} = f(\mathbf{x})$. In this case, the likelihood $p_X(\mathbf{x})$ can be expressed in terms of a base distribution $p_Z$ and the Jacobian determinant of $f$:

$$p_X(\mathbf{x}) = p_Z(\mathbf{z}) \left| \frac{\mathrm{d}\mathbf{z}}{\mathrm{d}\mathbf{x}} \right|, \tag{1}$$

where $p_Z$ is typically chosen to be a simple factorized distribution such as a Gaussian, and $f$ is a function with learnable parameters that is referred to as a flow. Drawing a sample $\mathbf{x} \sim p_X$ is equivalent to drawing a sample $\mathbf{z} \sim p_Z$ and computing $\mathbf{x} = f^{-1}(\mathbf{z})$.

### 2.1 The Matrix Exponential

The matrix exponential gives a method to construct an invertible matrix from any dimensionality preserving linear transformation. For any square (possibly non-invertible) matrix $\mathbf{M}$, the matrix exponential is given by the power series:

$$\exp(\mathbf{M}) \equiv \mathbf{I} + \frac{\mathbf{M}}{1!} + \frac{\mathbf{M}^2}{2!} + \ldots = \sum_{i=0}^{\infty} \frac{\mathbf{M}^i}{i!}. \tag{2}$$

The matrix exponential is well-defined as the series always converges. Additionally, the matrix exponential has two very useful properties: *i)* computing the inverse of the matrix exponential has the same computational complexity as the exponential itself, and *ii)* the determinant of the matrix exponential can be computed easily using the trace:

$$\exp(\mathbf{M})^{-1} = \exp(-\mathbf{M}) \quad \text{and} \quad \log\det[\exp(\mathbf{M})] = \operatorname{Tr}\mathbf{M}. \tag{3}$$

The matrix exponential has been largely used in the field of ODEs. Consider the linear ordinary differential equation $\frac{\mathrm{d}\mathbf{x}}{\mathrm{d}t} = \mathbf{M}\mathbf{x}$. Given the initial condition $\mathbf{x}(t = 0) = \mathbf{x}_0$, the solution for $\mathbf{x}(t)$ at time $t$ can be written using the matrix exponential: $\mathbf{x}(t) = \exp(\mathbf{M} \cdot t) \cdot \mathbf{x}_0$. As a result we can express the solution class of the matrix exponential: The matrix exponential can model any linear transformation that is the solution to a linear ODE. Note that not all invertible matrices can be expressed as an exponential, for instance matrices with negative determinant are not possible.

## 2.2 Convolutions as Matrix Multiplications

Convolutional layers in deep learning can be expressed as matrix multiplications. Let $\mathbf{m} \star \mathbf{x}$ denote a convolution[1], then there exists an equivalent matrix $\mathbf{M}$ such that the convolution is equivalent to the matrix multiplication $\mathbf{M}\vec{\mathbf{x}}$, where $\vec{\cdot}$ vectorizes $\mathbf{x}$. An example is provided in Figure 2. In these examples we use zero-padded convolutions, for periodic and reflective padded convolutions a slightly different equivalent matrix exists. An important detail to notice is that the equivalent matrix is typically unreasonably large to store in memory, its dimensions grow quadratically with the dimension of $\mathbf{x}$. For example, for 2d signals it has size $hwc \times hwc$, where $h$ is height, $w$ is width and $c$ denotes number of channels. In practice the equivalent matrix is never stored but instead, it is a useful tool to utilize concepts from linear algebra.

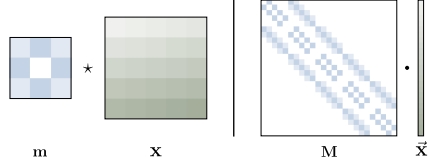

Figure 2: A convolution of a signal $\mathbf{x}$ with a kernel $\mathbf{m}$ (left) is equivalent to a matrix multiplication using a matrix $\mathbf{M}$ and a vectorized signal $\vec{\mathbf{x}}$ (right). In this example, $\mathbf{x}$ has a single channel with spatial dimensions $5 \times 5$. The convolution is zero-padded with one pixel on all sides. A white square indicates its value is zero.

## 3 The Convolution Exponential

We introduce a new method to build linear flows, by taking the exponential of a linear transformation. As the main example we take the exponential of a convolutional layer, which we name the *convolution exponential*. Since a convolutional is linear, it can be expressed as a matrix multiplication (section 2.2). For a convolution with a kernel $\mathbf{m}$, there exists an associated equivalent matrix using the matrix $\mathbf{M}$ such that $\mathbf{m} \star \mathbf{x}$ and $\mathbf{M} \cdot \vec{\mathbf{x}}$ are equivalent. We define the convolution exponential:

$$\mathbf{z} = \mathbf{m} \star_e \mathbf{x}, \tag{4}$$

for a kernel $\mathbf{m}$ and signal $\mathbf{x}$ as the output of the matrix exponential of the equivalent matrix: $\vec{\mathbf{z}} = \exp(\mathbf{M}) \cdot \vec{\mathbf{x}}$, where the difference between $\mathbf{z}$ and $\vec{\mathbf{z}}$ is a vectorization or *reshape* operation that can be easily inverted. Notice that although $\star$ is a linear operation with respect to $\mathbf{m}$ and $\mathbf{x}$, the exponential operation $\star_e$ is only linear with respect to $\mathbf{x}$. Using the properties of the matrix exponential, the inverse is given by $(-\mathbf{m}) \star_e \mathbf{x}$, and the log Jacobian determinant is the trace of $\mathbf{M}$. For a 2d convolutional layer the trace is $hw \cdot \sum_c m_{c,c,m_y,m_x}$ given the 4d kernel tensor $\mathbf{m}$, where height is $h$, width is $w$, the spatial center of the kernel is given by $m_y, m_x$ and $c$ iterates over channels. As an example, consider the convolution in Figure 2. The exponential of its equivalent matrix is depicted in Figure 1. In contrast with a standard convolution, the convolution exponential guaranteed to be invertible, and computing the Jacobian determinant is computationally cheap.

**Implicit iterative computation**
Due to the popularity of the matrix exponential as a solutions to ODEs, numerous methods to compute the matrix exponential with high numerical precision exist (Arioli et al., 1996; Moler and Van Loan, 2003). However, these methods typically rely on storing the matrix $\mathbf{M}$ in memory, which is very expensive for transformations such as convolutional layers. Instead, we propose to solve the exponential using matrix vector products $\mathbf{M}\vec{\mathbf{x}}$. The *exponential* matrix vector product $\exp(\mathbf{M})\vec{\mathbf{x}}$ can

**Algorithm 1** Implicit matrix exponential

   **Inputs:** $\mathbf{M}, \mathbf{x}$
   **Output:** $\mathbf{z}$
   let $\boldsymbol{\pi} \leftarrow \mathbf{x}, \mathbf{z} \leftarrow \mathbf{x}$
   **for** $i = 1, \ldots, T$ **do**
      $\boldsymbol{\pi} \leftarrow \mathbf{M} \cdot \boldsymbol{\pi}/i$
      $\mathbf{z} \leftarrow \mathbf{z} + \boldsymbol{\pi}$
   **end for**

**Algorithm 2** General linear exponential

   **Inputs:** $\mathbf{x}$, linear function $L : \mathcal{X} \to \mathcal{X}$
   **Output:** $\mathbf{z}$
   let $\boldsymbol{\pi} \leftarrow \mathbf{x}, \mathbf{z} \leftarrow \mathbf{x}$
   **for** $i = 1, \ldots, T$ **do**
      $\boldsymbol{\pi} \leftarrow L(\boldsymbol{\pi})/i$
      $\mathbf{z} \leftarrow \mathbf{z} + \boldsymbol{\pi}$
   **end for**

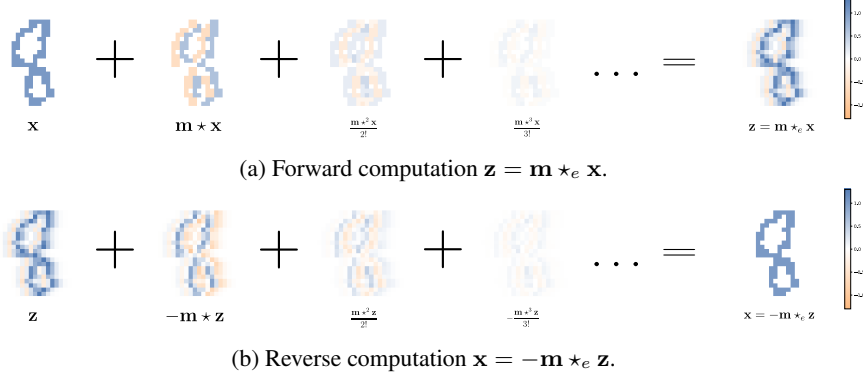

(a) Forward computation $\mathbf{z} = \mathbf{m} \star_e \mathbf{x}$.

(b) Reverse computation $\mathbf{x} = -\mathbf{m} \star_e \mathbf{z}$.

Figure 3: Visualization of the feature maps in the convolution exponential with the edge filter $\mathbf{m} = [0.6, 0, -0.6]$. Note that the notation $\mathbf{w} \star^2 \mathbf{x}$ simply means $\mathbf{w} \star (\mathbf{w} \star \mathbf{x})$, *that is* two subsequent convolutions on $\mathbf{x}$. Similarly for any $n$ the expression $\mathbf{w} \star^n \mathbf{x} = \mathbf{w} \star (\mathbf{w} \star^{n-1} \mathbf{x})$.

be computed implicitly using the power series, multiplied by any vector $\vec{\mathbf{x}}$ using only matrix-vector multiplications:

$$\exp(\mathbf{M}) \cdot \vec{\mathbf{x}} = \vec{\mathbf{x}} + \frac{\mathbf{M} \cdot \vec{\mathbf{x}}}{1!} + \frac{\mathbf{M}^2 \cdot \vec{\mathbf{x}}}{2!} + \ldots = \sum_{i=0}^{\infty} \frac{\mathbf{M}^i \cdot \vec{\mathbf{x}}}{i!}, \tag{5}$$

where the term $\mathbf{M}^2 \cdot \mathbf{x}$ can be expressed as two matrix vector multiplications $\mathbf{M}(\mathbf{M} \cdot \mathbf{x})$. Further, computation from previous terms can be efficiently re-used as described in Algorithm 1. Using this fact, the convolution exponential can be directly computed using the series:

$$\mathbf{m} \star_e \mathbf{x} = \mathbf{x} + \frac{\mathbf{m} \star \mathbf{x}}{1!} + \frac{\mathbf{m} \star (\mathbf{m} \star \mathbf{x})}{2!} + \ldots, \tag{6}$$

which can be done efficiently by simply setting $L(\mathbf{x}) = \mathbf{m} \star \mathbf{x}$ in Algorithm 2. A visual example of the implicit computation is presented in Figure 3.

**Power series convergence**

Even though the exponential can be solved implicitly, it is uncertain how many terms of the series will need to be expanded for accurate results. Moreover, it is also uncertain whether the series can be computed with high numerical precision. To resolve both issues, we constrain the induced matrix norm of the linear transformation. Given the $p$-norm on the matrix $\mathbf{M}$, a theoretical upper bound for the size of the terms in the power series can be computed using the inequality: $||\mathbf{M}^i \mathbf{x}||_p \leq ||\mathbf{M}||_p^i ||\mathbf{x}||_p$. Hence, an upper bound for relative size of the norm of a term at iteration $i$, is given by $||\mathbf{M}||_p^i / i!$. Notice that the factorial term in the denominator causes the exponential series to converges very fast, which is depicted in Figure 4.

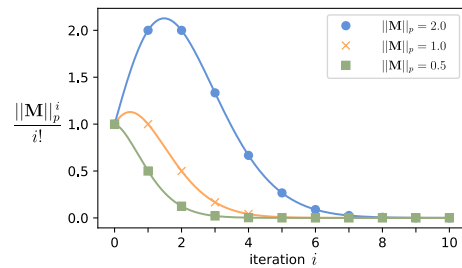

Figure 4: Upper bound of the norm of a term in the power series $||\mathbf{M}^i \mathbf{x}||_p / i!$ at iteration $i$, relative to the size of the input $||\mathbf{x}||_p$ given a matrix norm.

In our experiments we constrain $\mathbf{M}$ using spectral normalization (Miyato et al., 2018; Gouk et al., 2018), which constrains the $\ell_2$ norm of the matrix ($p = 2$) and can be computed efficiently for convolutional layers and standard linear layers. Even though the algorithm approximates the $\ell_2$ norm, in practice the bound is sufficiently close to produce convergence behaviour as shown in Figure 4. Moreover, the figure depicts worst-case behaviour given the norm, and typically the series converges far more rapidly. In experiments we normalize the convolutional layer using a $\ell_2$ coefficient of 0.9 and we find that expanding around 6 terms of the series is generally sufficient. An interesting byproduct is that the transformations that can be learned by the exponential will be limited to linear ODEs that are Lipschitz constraint. In cases where it is useful to be able to learn permutations over channels, this limitation can be relieved by combining the exponential with cheap (Housholder) $1 \times 1$ convolutional layers.

## 3.1 Graph Convolution Exponential

In this section we extend the Convolution Exponential to graph structured data. Given a graph $\mathcal{G} = (\mathcal{V}, \mathcal{E})$ with nodes $v \in \mathcal{V}$ and edges $e \in \mathcal{E}$. We define a matrix of nodes $\times$ features $\mathbf{X} \in \mathbb{R}^{N \times nf}$, an adjacency matrix $\mathbf{A} \in \mathbb{R}^{N \times N}$ and a degree matrix $D_{ii} = \sum_j A_{ij}$. Employing a similar notation as Kipf and Welling (2016), a linear graph convolutional layer GCL : $\mathbb{R}^{N \times nf} \to \mathbb{R}^{N \times nf}$ can be defined as:

$$\text{GCL}_\theta(\mathbf{X}) = \mathbf{IX}\boldsymbol{\theta}_0 + \mathbf{D}^{-\frac{1}{2}}\mathbf{A}\mathbf{D}^{-\frac{1}{2}}\mathbf{X}\boldsymbol{\theta}_1, \tag{7}$$

where $\boldsymbol{\theta}_0, \boldsymbol{\theta}_1 \in \mathbb{R}^{nf \times nf}$ are free parameters. Since the output in the graph convolution linearly depends on its inputs, it can also be expressed as a product of some equivalent matrix $\mathbf{M} \in \mathbb{R}^{N \cdot nf \times N \cdot nf}$ with a vectorized signal $\vec{\mathbf{X}} \in \mathbb{R}^{N \cdot nf}$. Note that the trace of this equivalent matrix Tr $\mathbf{M}$ is is equal to the trace of $\boldsymbol{\theta}_0$, multiplied by the number of nodes, *i.e.* Tr $\mathbf{M} = N$ Tr $\boldsymbol{\theta}_0$. This is because the adjacency matrix $\mathbf{A}$ contains zeros on its diagonal and all self-connections are parametrized by $\boldsymbol{\theta}_0$. The proofs to obtain $\mathbf{M}$ from equation 7 and its trace Tr $\mathbf{M}$ are shown in Appendix B.

The graph convolution exponential can be computed by replacing $L$ with the function GCL in Algorithm 2. Since the size and structure of graphs may vary, the norm $||\mathbf{M}||$ changes depending on this structure even if the parameters $\boldsymbol{\theta}_0$ and $\boldsymbol{\theta}_1$ remain unchanged. As a rule of thumb we find that the graph convolution exponential converges quickly when the norm $||\boldsymbol{\theta}_0||_2$ is constrained to one divided by the maximum number of neighbours, and $||\boldsymbol{\theta}_1||_2$ to one (as it is already normalized via $\mathbf{D}$).

## 3.2 General Linear Exponentials and Equivariance

In the previous section we generalized the exponential to convolutions and graph convolutions. Although these convolutional layers themselves are equivariant transformations (Cohen and Welling, 2016; Dieleman et al., 2016), it is unclear whether the exponentiation retains this property. In other words: do exponentiation and equivariance commute?

Equivariance under $K$ is defined as $[K, M] = KM - MK = 0$ where $M$ is a general transformation that maps one layer of the neural network to the next layer, which is in this case a convolutional layer. It states that first performing the map $M$ and then the symmetry transform $K$ is equal to first transforming with $K$ and then with $M$, or concisely $KM = MK$. Although the symmetry transformation in the input layer and the activation layer are the same (this is less general than the usual equivariance constraint which is of the form $K_1 M = M K_2$), this definition is however still very general and encompasses group convolutions (Cohen and Welling, 2016; Dieleman et al., 2016) and permutation equivariant graph convolutions (Maron et al., 2019). Below we show that indeed, the exponentation retains this form of equivariance.

**Theorem 1**: Let $M$ be a dimensionality preserving linear transformation. If $M$ is equivariant with respect to $K$ such that $[K, M] = 0$, then the exponential of $M$ is also equivariant with respect to $K$, meaning that $[K, \exp M] = 0$.

*Proof.* Since $[K, M] = 0$, we have that:

$$\begin{aligned} [K, MM] &= KMM - MMK = KMM - MKM + MKM - MMK \\ &= [K, M]M + M[K, M] = 0. \end{aligned} \tag{8}$$

From symmetry of the operator $[..., ...]$ and induction it follows that $[K^n, M^m] = 0$ for positive powers $n, m$. Moreover, any linear combination of any collection of powers commutes as well. To show that the exponential is equivariant, we define $\exp_n M$ as the truncated exponential taking only the first $n$ terms of the series. Then $[K, \exp M] = [K, \lim_{n \to \infty} \exp_n M] = \lim_{n \to \infty}[K, \exp_n M] = 0$, because each $[K, \exp_n M] = 0$ for any positive integer $n$ and $[..., ...]$ is continuous and thus preserves limits. This answers our question that indeed $[K, \exp M] = 0$ and thus the exponentiation of $M$ is also equivariant. For the interested reader, this result can also be more directly obtained from the relationship between Lie algebras and Lie groups.

# 4 Generalized Sylvester Flows

Sylvester Normalizing Flows (SNF) (van den Berg et al., 2018) takes advantage of the Sylvester identity $\det(\mathbf{I} + \mathbf{AB}) = \det(\mathbf{I} + \mathbf{BA})$ that allows to calculate a determinant of the transformation

$\mathbf{z} = \mathbf{x} + \mathbf{A}h(\mathbf{Bx} + \mathbf{b})$ in an efficient manner. Specifically, van den Berg et al. (2018) parametrize $\mathbf{A}$ and $\mathbf{B}$ using a composition of a shared orthogonal matrix $\mathbf{Q}$ and triangular matrices $\mathbf{R}$, $\tilde{\mathbf{R}}$ such that $\mathbf{A} = \mathbf{Q}^\mathrm{T}\tilde{\mathbf{R}}$ and $\mathbf{B} = \mathbf{RQ}$. However, the original Sylvester flows utilize fully connected parametrizations, and not convolutional ones. We introduce an extension of Sylvester flows which we call *generalized Sylvester flows*. The transformation is described by:

$$\mathbf{z} = \mathbf{x} + \mathbf{W}^{-1}f_{\mathrm{AR}}\left(\mathbf{Wx}\right), \tag{9}$$

where $\mathbf{W}$ can be any invertible matrix and $f_{\mathrm{AR}}$ is a smooth autoregressive function. In this case the determinant can be computed using:

$$\det\left(\frac{\mathrm{d}\mathbf{z}}{\mathrm{d}\mathbf{x}}\right) = \det\left(\mathbf{I} + \mathbf{J}_{f_{\mathrm{AR}}}(\mathbf{Wx})\mathbf{WW}^{-1}\right) = \det\left(\mathbf{I} + \mathbf{J}_{f_{\mathrm{AR}}}(\mathbf{Wx})\right), \tag{10}$$

where $\mathbf{J}_{f_{\mathrm{AR}}}(\mathbf{Wx})$ denotes the Jacobian of $f_{\mathrm{AR}}$, which is triangular because $f_{\mathrm{AR}}$ is autoregressive. We can show that *1)* generalized Sylvester flows are invertible and *2)* that they generalize the original Sylvester flows.

**Theorem 2**: Let $\mathbf{W}$ be an invertible matrix. Let $f_{\mathrm{AR}} : \mathbb{R}^d \to \mathbb{R}^d$ be a smooth autoregressive function (*i.e.*, $\frac{\partial f_{\mathrm{AR}}(x)_i}{\partial x_j} = 0$ if $j > i$). Additionally, constrain $\frac{\partial f_{\mathrm{AR}}(x)_i}{\partial x_i} > -1$. Then the transformation given by (9) is invertible.

*Proof.* The vectors of matrix $\mathbf{W}$ form a basis change for $\mathbf{x}$. Since the basis change is invertible, it suffices to show that the transformation in the new basis is invertible. Multiplying Equation 9 by $\mathbf{W}$ from the left gives:

$$\underbrace{\mathbf{Wz}}_{\mathbf{v}} = \underbrace{\mathbf{Wx}}_{\mathbf{u}} + f_{\mathrm{AR}}\big(\underbrace{\mathbf{Wx}}_{\mathbf{u}}\big), \tag{11}$$

The transformation $\mathbf{v} = \mathbf{u} + f_{\mathrm{AR}}(\mathbf{u})$ combines an identity function with an autoregressive function for which the diagonal of the Jacobian is strictly larger than $-1$. As a result, the entire transformation from $\mathbf{u}$ to $\mathbf{v}$ has a triangular Jacobian with diagonal values strictly larger than $0$ and is thus invertible (for a more detailed treatment see (Papamakarios et al., 2019). Since the transformation in the new basis from $\mathbf{u}$ to $\mathbf{v}$ is invertible, the transformation from $\mathbf{x}$ to $\mathbf{z}$ given in (9) is indeed also invertible.

**Theorem 3**: The original Sylvester flow $\mathbf{z} = \mathbf{x} + \mathbf{Q}^T\tilde{\mathbf{R}}h(\mathbf{RQx} + \mathbf{b})$, is a special case of the generalized Sylvester flow (9). *Proof: see Appendix A.*

In summary, Theorem 2 demonstrates that the generalized Sylvester Flow is invertible and Theorem 3 shows that the original Sylvester Flows can be modelled as a special case by this new transformation. The generalization admits the use of any invertible linear transformation for $\mathbf{W}$, such as a convolution exponential. In addition, it allows the use of general autoregressive functions.

### 4.1 Inverting Sylvester Flows

Recall that we require that the diagonal values of $\mathbf{J}_{f_{\mathrm{AR}}}$ are greater than $-1$. If we additionally constrain the maximum of this diagonal to $+1$, then the function becomes a one-dimensional contraction, given that the other dimensions are fixed. Using this, the inverse of Sylvester flows can be easily computed using a fixed point iteration. Firstly, compute $\mathbf{v} = \mathbf{Wz}$ and let $\mathbf{u}^{(0)} = \mathbf{v}$. At this point the triangular system $\mathbf{v} = \mathbf{u} + f_{\mathrm{AR}}(\mathbf{u})$ can be solved for $\mathbf{u}$ using the fixed-point iteration:

$$\mathbf{u}^{(t)} = \mathbf{v} - f_{\mathrm{AR}}(\mathbf{u}^{(t-1)}). \tag{12}$$

Subsequently, $\mathbf{x}$ can be obtained by computing $\mathbf{x} = \mathbf{W}^{-1}\mathbf{u}$. This procedure is valid both for our method and the original Sylvester flows. Although the fixed-point iteration is identical to (Behrmann et al., 2019), the reason that Sylvester flows converge is because *i)* the function $f_{\mathrm{AR}}$ is a contraction in one dimension, *ii)* the function is autoregressive (for a proof see Appendix A). The entire function $f_{\mathrm{AR}}$ does not need to be a contraction. Solving an autoregressive inverse using fixed-point iteration is generally faster than solving the system iteratively (Song et al., 2020; Wiggers and Hoogeboom, 2020).

Specifically, we choose that $f_{\mathrm{AR}}(\mathbf{u}) = \gamma \cdot s_2(\mathbf{u}) \odot \tanh\left(\mathbf{u} \odot s_1(\mathbf{u}) + t_1(\mathbf{u})\right) + t_2(\mathbf{u})$, where $s_1, s_2, t_1, t_2$ are *strictly* autoregressive functions parametrized by neural networks with a shared representation. Also $s_1, s_2$ utilize a final $\tanh$ function so that their output is in $(-1, 1)$ and $0 < \gamma < 1$, which we set to $0.5$. This transformation is somewhat similar to the construction of the original Sylvester flows (van den Berg et al., 2018), with the important difference that $s_1, s_2, t_1, t_2$ can now be modelled by any strictly autoregressive function.

## 4.2 Convolutional Sylvester Flows

Generalized Sylvester flows and the convolution exponential can be naturally combined to obtain *Convolutional Sylvester Flows* (CSFs). In Equation 9 we let $\mathbf{W} = \exp(\mathbf{M})\mathbf{Q}$, where $\mathbf{M}$ is the equivalent matrix of a convolution with filter $\mathbf{m}$. In addition $\mathbf{Q}$ is an orthogonal $1 \times 1$ convolution modeled by Householder reflections (Tomczak and Welling, 2016; Hoogeboom et al., 2019a):

$$\mathbf{z} = \mathbf{x} + \mathbf{Q}^{\mathrm{T}} \left( (-\mathbf{m}) \star_e f_{\mathrm{AR}} \left( \mathbf{m} \star_e \mathbf{Q}\mathbf{x} \right) \right), \qquad (13)$$

where the function $f_{\mathrm{AR}}$ is modelled using autoregressive convolutions (Germain et al., 2015; Kingma et al., 2016). For this transformation the determinant $\det \left( \frac{\mathrm{d}\mathbf{z}}{\mathrm{d}\mathbf{x}} \right) = \det \left( \mathbf{I} + \mathbf{J}_{f_{\mathrm{AR}}} \left( \mathbf{m} \star_e \mathbf{Q}\mathbf{x} \right) \right)$, which is straightforward to compute as $\mathbf{J}_{f_{\mathrm{AR}}}$ is triangular.

## 5 Related Work

Deep generative models can be broadly divided in likelihood based model such as autoregressive models (ARMs) (Germain et al., 2015), Variational AutoEncoders (VAEs) (Kingma and Welling, 2014), Normalizing flows (Rezende and Mohamed, 2015), and adversarial methods (Goodfellow et al., 2014). Normalizing flows are particularly attractive because they admit exact likelihood estimation and can be designed for fast sampling. Several works have studied equivariance in flow-based models Köhler et al. (2019); Rezende et al. (2019).

Linear flows are generally used to mix information in-between triangular maps. Existing transformations in literature are permutations (Dinh et al., 2017), orthogonal transformations Tomczak and Welling (2016); Goliński et al. (2019), $1 \times 1$ convolutions (Kingma and Dhariwal, 2018), low-rank Woodbury transformations (Lu and Huang, 2020), emerging convolutions (Hoogeboom et al., 2019a), and periodic convolutions (Finzi et al., 2019; Karami et al., 2019; Hoogeboom et al., 2019a). From these transformations only periodic and emerging convolutions have a convolutional parametrization. However, periodicity is generally not a good inductive bias for images, and since emerging convolutions are autoregressive, their inverse requires the solution to an iterative problem. Notice that Goliński et al. (2019) utilize the matrix exponential to construct orthogonal transformations. However, their method cannot be utilized for convolutional transformations since they compute the exponential matrix explicitly. Our linear exponential can also be seen as a linear neural ODE (Chen et al., 2018), but the methods are used for different purposes and are computed differently. In Li et al. (2019) learn approximately orthogonal convolutional layers to prevent Lipschitz attenuation, but these cannot be straightforwardly applied to normalizing flows without stricter guarantees on the approximation.

There exist many triangular flows in the literature such as coupling layers (Dinh et al., 2015, 2017), autoregressive flows (Germain et al., 2015; Kingma et al., 2016; Papamakarios et al., 2017; Chen et al., 2018; De Cao et al., 2019; Song et al., 2019; Nielsen and Winther, 2020), spline flows (Durkan et al., 2019b,a) and polynomial flows (Jaini et al., 2019). Other flows such as Sylvester Flows (van den Berg et al., 2018) and Residual Flows (Behrmann et al., 2019; Chen et al., 2019) learn invertible *residual* transformations. Sylvester Flows ensure invertibility by orthogonal basis changes and constraints on triangular matrices. Our interpretation connects Sylvester Flows to more general triangular functions, such as the ones described above. Residual Flows ensure invertibility by constraining the Lipschitz continuity of the residual function. A disadvantage of residual flows is that computing the log determinant is not exact and the power series converges at a slower rate than the exponential.

## 6 Experiments

Because image data needs to be dequantized Theis et al. (2016); Ho et al. (2019), we optimize the expected lowerbound (ELBO) of the log-likelihood. The performance is compared in terms of negative ELBO and negative log-likelihood (NLL) which is approximated with 1000 importance weighting samples. Values are reported in bits per dimension on CIFAR10. The underlying matrix for the exponential is initialized with unusually small values such that the initial exponent will approximately yield an identity transformation. See Appendix A for a comparison of Sylvester flows when used to model the variational distribution in VAEs.

## 6.1 Mixing for generative flows

In this experiment the convolution exponential is utilized as a linear layer in-between affine coupling layers. For a fair comparison, all the methods are implemented in the same framework, and are optimized using the same procedure. For details regarding architecture and optimization see Appendix C. The convolution exponential is compared to other linear mixing layers from literature: $1 \times 1$ convolutions (Kingma and Dhariwal, 2018), emerging convolutions (Hoogeboom et al., 2019a), and Woodbury transformations Lu and Huang (2020). The number of intermediate channels in the coupling layers are adjusted slightly such that each method has an approximately equal parameter budget. The experiments show that our method outperforms all other methods measured in negative ELBO and log-likelihood (see Table 1). The timing experiments are run using four NVIDIA GTX 1080Ti GPUs for training and a single GPU for sampling. Interestingly, even though emerging convolutions also have a convolutional parametrization, their performance is worse than the convolution exponential. This indicates that the autoregressive factorization of emerging convolutions somewhat limits their flexibility, and the exponential parametrization works better.

Table 1: Generative modelling performance with a generative flow. Results computed using $\log_2$ averaged over dimensions, i.e. bits per dimension. Results were obtained by re-implementing the relevant method in the same framework for a fair comparison. Models have an approximately equal parameter budget.

| Mixing type | CIFAR10 | | Runtime (%) | |
| --- | --- | --- | --- | --- |
| | -ELBO | NLL | Training | Sampling |
| $1 \times 1$ (Kingma and Dhariwal, 2018) | $3.285 \pm 0.008$ | $3.266 \pm 0.007$ | 100.0% | 100.0% |
| Emerging (Hoogeboom et al., 2019a) | $3.245 \pm 0.002$ | $3.226 \pm 0.002$ | 103.2% | 1223.5% |
| Woodbury (Lu and Huang, 2020) | $3.247 \pm 0.003$ | $3.228 \pm 0.003$ | 133.2% | 135.4% |
| Convolution Exponential | $\mathbf{3.237} \pm 0.002$ | $\mathbf{3.218} \pm 0.003$ | 104.6% | 115.8% |

## 6.2 Density modelling using residual transformations

Since Sylvester Flows are designed to have a residual connection, it is natural to compare their performance to invertible residual networks (Behrmann et al., 2019) which were improved to have unbiased log determinant estimates, and subsequently named residual flows (Chen et al., 2019). For a fair comparison, we run the code from (Chen et al., 2019) inside our framework using the same architecture and number of optimizer steps. For reference we also train a typical coupling-based flow with the same architecture. For more details please refer to Appendix C. Note that a direct comparison to the results in Table 1 may not be fair, as the network architectures are structurally different. The results show that Sylvester flows considerably outperform residual networks in image density estimation. Additionally, the memory footprint during training of residual blocks is roughly twice of the other models, due to the the Jacobian determinant estimation. When correcting for this, the equal memory budget result is obtained. In this case, the residual block flow is even

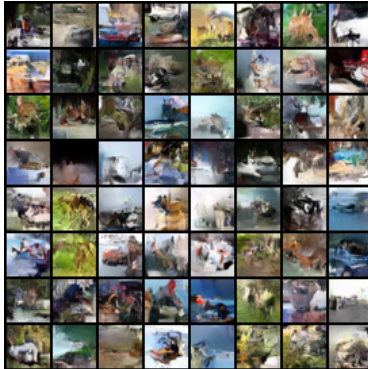

Figure 5: Samples from a generative Convolutional Sylvester flow trained on CIFAR10.

outperformed by a coupling flow. We hypothesize that this is caused by the strict Lipschitz continuity that has to be enforced for residual flows. The ablation study in Table 3 shows the effect of using

Table 2: Invertible Residual Networks as density model. Results for residual flows were obtained by running the residual block code from (Chen et al., 2019) in our framework.

| Model | Unif. deq. | | Var. deq. | |
| --- | --- | --- | --- | --- |
| | -ELBO | NLL | -ELBO | NLL |
| Baseline Coupling Flow | 3.38 | 3.35 | 3.27 | 3.25 |
| Residual Block Flows | 3.37 | - | 3.26 | - |
| with equal memory budget | 3.44 | - | 3.35 | - |
| Convolutional Sylvester Flows | **3.32** | **3.29** | **3.21** | **3.19** |

Table 3: Ablation studies: a study of the effect of the generalization, and the basis change.

| Model | CIFAR10 | |
| --- | --- | --- |
| | -ELBO | NLL |
| Conv. Sylvester | **3.21** | **3.19** |
| without $f_{AR}$ | 3.44 | 3.42 |
| without basis | 3.27 | 3.25 |

non-generalized Sylvester Flows, and the effect of not doing the basis change. Since the original Sylvester Flows (van den Berg et al., 2018) are not convolutional, it is difficult to directly compare these methods. The ablation result without the generalization using $f_{\text{AR}}$, is the closest to a convolutional interpretation of the original Sylvester flows, although it already has the added benefit of the exponential basis change. Even so, our Convolutional Sylvester flows considerably outperform this non-generalized Sylvester flow.

### 6.3 Graph Normalizing Flows

In this section we compare our Graph Convolution Exponential with other methods from the literature. As a first baseline we use a baseline coupling flow Dinh et al. (2017) that does not exploit the graph structure of the data. The second baseline is a Graph Normalizing Flows that uses graph coupling layers as described in (Liu et al., 2019). Since normalizing flows for edges of the graph is an open problem, following Liu et al. (2019) we assume a fully connected adjacency matrix. Our method then adds a graph convolution exponential layer preceding every coupling layer. For further implementation details refer to Appendix B. Following Liu et al. (2019) we test the methods on the graph datasets Mixture of Gaussian (MoG) and Mixture of Gaussians Ring (MoG-Ring), which are essentially mixtures of permutation of Gaussians. The original MoG dataset considers 4 Gaussians, which we extend to 9 and 16 Gaussians obtaining two new datasets MoG-9 and MoG-16 to study performance when the number of nodes increase. The MoG-Ring entropy is estimated using importance weighting to marginalize over the rotation. Results are presented in Table 4. Adding the graph convolution exponential improves the performance in all four datasets. The improvement becomes larger as the number of nodes increases (e.g. MoG-9 and MoG-16), which is coherent with the intuition that our Graph Convolution Exponential propagates information among nodes in the mixing layer.

Table 4: Benchmark over different models for synthetic graph datasets. Per-node Negative Log Likelihood (NLL) is reported in nats.

| Model | MoG-4 | MoG-9 | MoG-16 | MoG-Ring |
|---|---|---|---|---|
| Dataset entropy | 3.63 ±0.000 | 4.26 ±0.013 | - | ≤4.05 ±0.001 |
| Baseline Coupling Flow | 3.89 ±0.012 | 6.14 ±0.012 | 7.20 ±0.021 | 4.35 ±0.023 |
| Graph Normalizing Flow | 3.69 ±0.016 | 4.60 ±0.067 | 5.38 ±0.048 | 4.22 ±0.025 |
| with Graph Convolution Exponential | **3.68** ±0.017 | **4.52** ±0.047 | **5.26** ±0.047 | **4.19** ±0.036 |

## 7  Conclusion

In this paper we introduced a new simple method to construct invertible transformations, by taking the *exponential* of any linear transformation. Unlike prior work, we observe that the exponential can be computed *implicitly*. Using this we developed new invertible transformations named *convolution exponentials* and *graph convolution exponentials*, and showed that they retain their equivariance properties under exponentiation. In addition, we generalize Sylvester Flows and propose *Convolutional Sylvester Flows*.

## Broader Impact

This paper discusses methods to improve the flexibility of normalizing flows, a method to learn high-dimensional distributions. Methods based on our work could potentially be used to generate realistic looking photographs. On the other hand, distribution modelling could also be used for fraud detection via outlier detection, which could help detect generated media. In summary, we believe this method may be somewhat distant from direct applications, but a future derived method could be used in the above mentioned scenarios.

## Funding Disclosure

There are no additional sources of funding to disclose.

## Footnotes

[1]In frameworks, convolutions are typically implemented as *cross-correlations*. We follow literature convention and refer to them as *convolutions* in text. In equations $\star$ denotes a cross-correlation and $\ast$ a convolution.

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
