[Supplementary Material]

# Supplementary Material for: The Convolution Exponential and Generalized Sylvester Flows

**Emiel Hoogeboom**
UvA-Bosch Delta Lab
University of Amsterdam
The Netherlands
e.hoogeboom@uva.nl

**Victor Garcia Satorras**
UvA-Bosch Delta Lab
University of Amsterdam
The Netherlands
v.garciasatorras@uva.nl

**Jakub M. Tomczak**
Vrije Universiteit Amsterdam
The Netherlands
j.m.tomczak@vu.nl

**Max Welling**
UvA-Bosch Delta Lab
University of Amsterdam
The Netherlands
m.welling@uva.nl

## A    Details for Generalized Sylvester Flows

Recall that the Generalized Sylvester Flows transformation is described by:

$$\mathbf{z} = \mathbf{x} + \mathbf{W}^{-1} f_{\mathrm{AR}}\left(\mathbf{Wx}\right). \tag{1}$$

**Theorem 3**: The original Sylvester flow $\mathbf{z} = \mathbf{x} + \mathbf{Q}^T \tilde{\mathbf{R}} h(\mathbf{RQx} + \mathbf{b})$, is a special case of the generalized Sylvester flow (1).

*Proof.* Let $\mathbf{W} = \mathbf{Q}$ and let $f_{\mathrm{AR}}(\mathbf{x}) = \tilde{\mathbf{R}} h(\mathbf{Rx} + \mathbf{b})$. Indeed, any orthogonal matrix is invertible, so $\mathbf{Q}$ can be modelled by $\mathbf{W}$. Also, note that $\mathbf{R}$ and $\tilde{\mathbf{R}}$ are upper triangular and $h$ is an elementwise function. The matrix product of the Jacobians is triangular, and thus $\tilde{\mathbf{R}} h(\mathbf{Rx} + \mathbf{b})$ has a triangular Jacobian and is therefore autoregressive. Hence, it can be modelled by $f_{\mathrm{AR}}$. Further, note that [10] bound $R_{ii} \tilde{R}_{ii} > \frac{-1}{||h'||_\infty}$, which ensures that the constraint $\frac{\partial f_{\mathrm{AR}}(\mathbf{x})_i}{\partial x_i} > -1$ is satisfied. Hence, $\mathbf{z} = \mathbf{x} + \mathbf{Q}^T \tilde{\mathbf{R}} h(\mathbf{RQx} + \mathbf{b})$ can be written as Equation 1 when writing $f_{\mathrm{AR}}(\mathbf{x}) = \tilde{\mathbf{R}} h(\mathbf{Rx} + \mathbf{b})$ and $\mathbf{M} = \mathbf{Q}$ without violating any constraints on $\mathbf{M}$ and $f_{\mathrm{AR}}$, and is therefore a special case.

**Remark 1:**
The increased expressivity originates from $f_{\mathrm{AR}}$ and not from $\mathbf{W}$. To see why, suppose we replace $\mathbf{Q}$ and $\mathbf{Q}^T$ in the original formulation by $\mathbf{W}$ and $\mathbf{W}^{-1}$. Consider that any real square matrix $\mathbf{W}$ may be decomposed as $\mathbf{Q_W R_W}$. Hence, compositions $\mathbf{W}^{-1}\tilde{\mathbf{R}}$ and $\mathbf{RW}$ can be written as $\mathbf{Q}_{\mathrm{W}}^{\mathrm{T}}\tilde{\mathbf{R}}'$ and $\mathbf{R}'\mathbf{Q_W}$, where $\tilde{\mathbf{R}}' = \mathbf{R}_{\mathrm{W}}^{-1}\tilde{\mathbf{R}}$ and $\mathbf{R}' = \mathbf{RR_W}$ which are both still upper triangular. Hence, we have shown that even if the orthogonal matrix $\mathbf{Q}$ is replaced by an invertible matrix $\mathbf{W}$, the transformation can still be written in terms of a shared orthogonal matrix $\mathbf{Q_W}$ and upper triangular matrices $\tilde{\mathbf{R}}'$ and $\mathbf{R}'$. Therefore, the source of the increased expressivity is not the replacement of $\mathbf{Q}$ by $\mathbf{W}$.

**Remark 2:**
Sylvester flows can also be viewed from a different perspective as a composition of three invertible transformations: a basis change, a residual invertible function and the inverse basis change. Specifically, let $f$ be an invertible function that can be written as $f(\mathbf{x}) = \mathbf{x} + g(\mathbf{x})$ (for instance $g$ can be the autoregressive function $f_{\mathrm{AR}}$ from above). Now apply a linear basis change $\mathbf{W}$ on $\mathbf{x}$, and the inverse on the output $f(\mathbf{x})$. Then $\mathbf{W}^{-1}f(\mathbf{Wx}) = \mathbf{W}^{-1}\mathbf{Wx} + \mathbf{W}^{-1}g(\mathbf{Wx}) = \mathbf{x} + \mathbf{W}^{-1}g(\mathbf{Wx})$. In other words, because the basis change is linear it distributes over addition and cancels in the identity connection, which results in a residual transformation.

The reason that we still utilize an invertible matrix $\mathbf{W}$ is that it allows more freedom when modelling using the convolution exponential (a $QR$ decomposition for convolutions can generally not be expressed in terms of convolutions). For fully connected settings $\mathbf{W}$ can safely be chosen to be orthogonal.

**Inverting Sylvester Flows**

The inverse of Sylvester flows can be easily computed using a fixed point iteration. Firstly, compute $\mathbf{v} = \mathbf{W}\mathbf{z}$ and let $\mathbf{u}^{(0)} = \mathbf{v}$. At this point the triangular system $\mathbf{v} = \mathbf{u} + f_{\mathrm{AR}}(\mathbf{u})$ can be solved for $\mathbf{u}$ using the fixed-point iteration:

$$\mathbf{u}^{(t)} = \mathbf{v} - f_{\mathrm{AR}}(\mathbf{u}^{(t-1)}). \tag{2}$$

To show that it converges, recall that we constrain the diagonal values of $\mathbf{J}_{f_{\mathrm{AR}}}$ to be greater than $-1$ and less than $+1$. In addition, we require $f_{\mathrm{AR}}$ to be Lipschitz continuous for some arbitrarily large value $L \in \mathbb{R}$. Note that since neural network are generally composed of linear layers and activation functions that are Lipschitz continuous, these networks themselves are also Lipschitz continuous. Note that although the function defined in section 4.1 has products which in theory do not have to be Lipschitz, in practice the function is used on bounded domains which makes $f_{\mathrm{AR}}$ Lipschitz continuous trivially. For a more theoretically rigorous function, values of $\mathbf{u}$ can be simply clipped beyond certain thresholds.

Firstly note that $|\frac{\partial f_{\mathrm{AR}}(u)_i}{\partial u_i}| < 1$. Moreover, by the construction of $f_{\mathrm{AR}}$ in section 4.1, the magnitude of the values on the diagonal of the Jacobian is bounded by the hyperparameter $\gamma$, so that $|\frac{\partial f_{\mathrm{AR}}(u)_i}{\partial u_i}| < \gamma$, where $0 \leq \gamma < 1$. Since the function is autoregressive, when all preceding dimensions are fixed, the function can be seen as a (one-dimensional) contraction.

We will show inductively over dimensions that the fixed point iteration for $\mathbf{u}^{(t)}$ converges. For the base case in the first dimension, note that $|u_1^{(t)} - u_1^{(t+1)}| \leq \gamma^t |u_1^{(0)} - u_1^{(1)}|$ and hence $u_1$ converges at a rate of $\gamma^t$. For the remainder of this proof we use the $\ell_1$ distance as distance metric.

For the induction step (for the higher dimensions), assume that $\mathbf{u}_{:n-1}$ converges at a rate of $t^{n-1}\gamma^t$, that is $||\mathbf{u}_{:n-1}^{(t-1)} - \mathbf{u}_{:n-1}^{(t)}|| \leq Ct^{n-1}\gamma^t$ for some constant $C \in \mathbb{R}$. Then $\mathbf{u}_{:n}$ converges at a rate of $t^n\gamma^t$. We can bound the difference for $u_n$ in dimension $n$ recursively using the Lipschitz continuity $L$ and bound on the diagonal of the Jacobian $\gamma$:

$$|u_n^{(t)} - u_n^{(t+1)}| \leq \gamma |u_n^{(t-1)} - u_n^{(t)}| + L||\mathbf{u}_{1:n-1}^{(t-1)} - \mathbf{u}_{1:n-1}^{(t)}||. \tag{3}$$

When expanding this equation and using that $||\mathbf{u}_{1:n-1}^{(t-1)} - \mathbf{u}_{1:n-1}^{(t)}|| \leq Ct^{n-1}\gamma^t$, we can write:

$$|u_n^{(t)} - u_n^{(t+1)}| \leq \gamma^t |u_n^{(0)} - u_n^{(1)}| + \sum_{t'=1}^{t} \gamma^{(t-t')} LCt'^{n-1}\gamma^{t'},$$

$$\leq \gamma^t |u_n^{(0)} - u_n^{(1)}| + t^n \gamma^t LC,$$

which is guaranteed to converge at least at a rate of $t^n\gamma^t$. The last inequality follows because $t^n \geq \sum_{t'=1}^{t} t'^{n-1}$. Since $\mathbf{u}_{1:n-1}$ converges already at a rate of $t^{n-1}\gamma^t$, the convergence rate for $\mathbf{u}_{1:n}$ is bounded by $t^n\gamma^t$. Combining this result with the base case, $\mathbf{u}_{1:1}$ converges with a rate of $\gamma^t$, gives the result that the convergence rate of the entire vector $\mathbf{u}$ is bounded by $t^{d-1}\gamma^t$, where $d$ is the dimensionality of $\mathbf{u}$. Studying this equation, we can recognize two factors that influence the convergence that we can easily control: The distance of outputs with respect to the distance of inputs in $f_{\mathrm{AR}}$, and the one-dimensional continuity $\gamma$. We find experimentally that constraining the Lipschitz continuity of the convolutional layers in $f_{\mathrm{AR}}$ to 1.5, and setting $\gamma = 0.5$ generally allows the fixed point iteration to converge within 50 iterations when using an absolute tolerance of $10^{-4}$.

## B   Graph Convolution Exponential

Given the product of three matrices $ABC$ with dimensions $k \times l$, $l \times m$ and $m \times n$ respectively we can express its vectorized form in the following way:

$$\mathrm{vec}(ABC) = \left(C^{\mathrm{T}} \otimes A\right) \mathrm{vec}(B) \tag{4}$$

Where $\otimes$ stands for the Kronecker product. We obtain the vectorized form of the Linear Graph Convolutional Layer by applying the above mentioned equation 4 to the graph convolutional layer equation as follows:

$$\text{vec}(\mathbf{IX}\boldsymbol{\theta}_0 + \mathbf{D}^{-\frac{1}{2}}\mathbf{AD}^{-\frac{1}{2}}\mathbf{X}\boldsymbol{\theta}_1) =$$
$$\text{vec}(\mathbf{IX}\boldsymbol{\theta}_0) + \text{vec}(\mathbf{D}^{-\frac{1}{2}}\mathbf{AD}^{-\frac{1}{2}}\mathbf{X}\boldsymbol{\theta}_1) =$$
$$(\boldsymbol{\theta}_0^T \otimes \mathbf{I})\,\text{vec}(\mathbf{X}) + (\boldsymbol{\theta}_1^T \otimes \mathbf{D}^{-\frac{1}{2}}\mathbf{AD}^{-\frac{1}{2}})\,\text{vec}(\mathbf{X}) =$$
$$(\boldsymbol{\theta}_0^T \otimes \mathbf{I} + \boldsymbol{\theta}_1^T \otimes \mathbf{D}^{-\frac{1}{2}}\mathbf{AD}^{-\frac{1}{2}})\,\text{vec}(\mathbf{X}) =$$
$$\mathbf{M}\,\text{vec}(\mathbf{X})$$

Now that we have analytically obtained $\mathbf{M}$ we can compute its trace by making use of the following Kronecker product property: $\text{tr}(\mathbf{A} \otimes \mathbf{B}) = \text{Tr}\,\mathbf{A}\,\text{Tr}\,\mathbf{B}$. The trace of $\mathbf{M}$ will be:

$$\text{Tr}\,(\mathbf{M}) = \text{Tr}(\boldsymbol{\theta}_0^T \otimes \mathbf{I} + \boldsymbol{\theta}_1^T \otimes \mathbf{D}^{-\frac{1}{2}}\mathbf{AD}^{-\frac{1}{2}}) =$$
$$\text{Tr}(\boldsymbol{\theta}_0^T)\,\text{Tr}(\mathbf{I}) + \text{Tr}(\boldsymbol{\theta}_1^T)\,\text{Tr}(\mathbf{D}^{-\frac{1}{2}}\mathbf{AD}^{-\frac{1}{2}}) =$$
$$\text{Tr}(\boldsymbol{\theta}_0^T)\,N + \text{Tr}(\boldsymbol{\theta}_1^T)\,0 =$$
$$N\,\text{Tr}(\boldsymbol{\theta}_0)$$

## C   Experimental Details

We train on the first 40000 images of CIFAR10, using the remaining 10000 for validation. The final performance is shown on the conventional 10000 test images.

### C.1   Mixing experiment

The flow architecture is multi-scale following [7]: Each level starts with a squeeze operation, and then 10 subflows which each consist of a linear mixing layer and an affine coupling layer [1]. The coupling architecture utilizes densenets as described in [3]. Further, we use variational dequantization [2], using the same flow architecture as for the density estimation, but using less subflows. Following [1, 7] after each level (except the final level) half the variables are transformed by another coupling layer and then factored-out. The final base distribution $p_Z$ is a diagonal Gaussian with mean and standard deviation. All methods are optimized using a batch size of 256 using the Adam optimizer [4] with a learning rate of 0.001 with standard settings. More details are given in Table 1. Notice that convexp mixing utilizes a convolution exponential and a $1 \times 1$ convolutions, as it tends to map close to the identity by the construction of the power series. Results are obtained by running models three times after random weight initialization, and the mean of the values is reported. Runs require approximately four to five days to complete. Results are obtained by running on four NVIDIA GeForce GTX 1080Ti GPUs, CUDA Version: 10.1.

Table 1: Architecture settings and optimization settings for the mixing experiments.

| Model | levels | subflows | epochs | lr decay | densenet depth | densenet growth | deq. levels | deq. subflows |
|---|---|---|---|---|---|---|---|---|
| $1 \times 1$ | 2 | 10 | 1000 | 0.995 | 8 | 64 | 1 | 4 |
| Emerging | 2 | 10 | 1000 | 0.995 | 8 | 63 | 1 | 4 |
| Woodbury | 2 | 10 | 1000 | 0.995 | 8 | 63 | 1 | 4 |
| ConvExp | 2 | 10 | 1000 | 0.995 | 8 | 63 | 1 | 4 |

### C.2   Invertible Residual Transformations experiment

The setup is identical to section C.1, where a single subflow is now either a residual block or a convolutional Sylvester flow transformation, with a leading actnorm layer [7]. The network architectures *inside* the Sylvester and residual network architectures all consist of three standard convolutional layers: A $3 \times 3$ convolution, a $1 \times 1$ convolution and another $3 \times 3$ convolution. These provide the translation and scale parameters for the Sylvester transformation, and they model the

Table 2: Architecture settings and optimization settings for the residual experiments. Dequantization (deq.) settings are not used for uniform dequantization.

| Model | levels | subflows | epochs | lr decay | channels | (deq. levels) | (deq. subflows) |
|---|---|---|---|---|---|---|---|
| Baseline Coupling | 2 | 20 | 1000 | 0.995 | 528 | 1 | 4 |
| Residual Block Flow | 2 | 20 | 1000 | 0.995 | 528 | 1 | 4 |
|    equal memory | 2 | 10 | 1000 | 0.995 | 528 | 1 | 4 |
| Conv. Sylvester | 2 | 20 | 1000 | 0.995 | 528 | 1 | 4 |

Table 3: The performance of VAEs with different normalizing flows as encoder distributions. Results are obtained by running three times with random initialization. In line with literature, binary MNIST is reported in nats and CIFAR10 is reported in bits per dimension.

| Model | bMNIST | | CIFAR10 | |
|---|---|---|---|---|
| | -ELBO | $-\log P(x)$ | -ELBO | $-\log P(x)$ |
| Gaussian [6] | 85.54 ±0.08 | 81.77 ±0.05 | 4.59 ±0.016 | 4.54 ±0.016 |
| Planar [8] | 85.65 ±0.18 | 81.79 ±0.08 | 4.59 ±0.008 | 4.55 ±0.007 |
| IAF [5] | 84.00 ±0.07 | 80.77 ±0.02 | 4.57 ±0.003 | 4.53 ±0.002 |
| H-SNF [10] | **83.34** ±0.06 | **80.38** ±0.02 | 4.58 ±0.001 | 4.54 ±0.001 |
| Generalized Sylvester (ours) | **83.29** ±0.04 | **80.41** ±0.02 | 4.57 ±0.009 | 4.54 ±0.009 |
| Conv. Gaussian | 87.41 ±0.06 | 83.12 ±0.03 | 3.90 ±0.010 | 3.82 ±0.011 |
| Conv. IAF [5] | 83.82 ±0.09 | 81.01 ±0.08 | 3.69 ±0.011 | 3.64 ±0.011 |
| Conv. SNF (ours) | 83.68 ±0.11 | 80.96 ±0.07 | **3.48** ±0.005 | **3.44** ±0.005 |

residual for the residual flows. These convolutions map to $528$ channels internally, where the first and last convolutional layers map to the respective input and output sizes. Note that for the coupling flow an important difference that a subflow consists of a coupling layer *and* a $1 \times 1$ convolution, because the coupling layer itself cannot mix information. All methods use the same dequantization flow and splitprior architecture that were described in section C.1. All methods are optimized using a batch size of 256 using the Adam optimizer [4] with a learning rate of $0.001$ with $\beta_1, \beta_2 = (0.9, 0.99)$. Results are obtained by running models a single after random weight initialization. More details are given in Table 2. Results are obtained by running on two NVIDIA GeForce GTX 1080Ti GPUs, CUDA Version: 10.1. Runs require approximately four to five days to complete. The residual block flow utilizes four GPUs as it requires more memory.

### C.3 Graph Normalizing Flow experiment

The normalizing flows in the graph experiments all utilize three subflows, where a subflow consists of an actnorm layer [7], a $1 \times 1$ convolution and an affine coupling layer [1]. In the model that utilizes the graph convolution exponential, the convolution exponential precedes each coupling layer. In the baseline coupling flow, the neural networks inside the coupling layers are 4-layer Multi Layer Perceptrons (MLPs) with Leaky Relu activations. In the graph normalizing flow, the neural networks inside the coupling layers are graph neural networks where node and edge operations are performed by a 2-layer and a 3-layer MLPs respectively with ReLU activations. All above mentioned neural networks utilize 64 hidden features.

All experiments are optimized for $35,000$ iterations, using a batch size of 256 and a starting learning rate of $2^{-4}$ with a learning rate decay factor of 0.1 every $15,000$ iterations. For testing we used $1,280,000$ samples, i.e. $5.000$ iterations with a batch size of 256.

## D   Variational posterior modelling in VAEs

This experiment utilizes normalizing flows as the variational posterior for a Variational AutoEncoder (VAE). The code is built upon the original Sylvester flow implementation [10] which contains a VAE with a single latent representation with a standard Gaussian prior. There are two differences: For CIFAR10 a discretized mixture of logistics [9] is used as output distribution for the decoder. Additionally, the gated convolutions are replaced by denseblock layers.

The proposed Convolutional Sylvester Flows outperform the other methods considerably in terms of ELBO and log-likelihood on CIFAR10. Interestingly, although the Convolutional Sylvester Flow outperforms other convolutional methods, the experiment shows that fully connected flows actually perform better on binary MNIST. Noteworthy for the comparison between the original Householder Sylvester flows (H-SNF) and our method is that H-SNF has four times more parameters than our convolutional Sylvester flows.