[Reviews · NeurIPS 2020]

Review 1

Summary and Contributions: The paper first provides a method for iteratively and implicitly computing the exponential of a linear transformation, which is guaranteed to be invertible, and whose inverse can be computed in the same iterative fashion, as well as allowing for efficient log det Jacobian computation in terms of the trace of the corresponding matrix operation. The method is applied to both traditional image convolutions as well as graph convolutions. The paper then presents a generalization of Sylvester normalizing flows (although the reliance on Sylvester's identity is maybe unclear) in terms of autoregressive residual transformations with a particular form. Finally, the paper combines the convolutional exponential and autoregressive residual transformations to propose a new flow transformation.

Strengths: Exploiting the exponential map and its useful properties (inverse, determinant) to parameterize an invertible linear transformation has been considered in the past, but the implicit and iterative computation presented here lends itself to more involved transformations such as convolutions, where the equivalent matrix operation would introduce prohibitive memory costs. The considerations of the power series convergence, and the inclusion of Figure 3 help convince the reader that the proposed series truncation can be both stable and accurate. I also appreciated the applications of the exponential in both traditional image convolutions, as well as graph convolutions, and the empirical evaluation in both domains. Overall the paper makes good use of informative figures (1, 2, 4).

Weaknesses: **Section 3.2**, which seeks to demonstrate that exponentiation preserves the equivariance of the underlying convolutions, is maybe a bit vague and difficult to understand. - 138-140: The terminology here is sudden and not defined anywhere in the paper e.g. what is a feature field, what is a capsule, why are pixels indexed by a single integer rather than a pair? Also 'x_i is the ith pixel at position x_i' should be 'x_i is the ith pixel at position i'. - There might be an easier way to prove commutativity of the exponential with equivariance using the general definition of the exponential map in terms of Lie algebras and Lie groups, but then again maybe it's better not to drag in that machinery if it's not needed. I'm just slightly hesitant of statements like 151-153 where properties holding for finite linear combinations are extended to infinite linear combinations. - Moreover, a more formal derivation would generally be more useful in 151-155. Despite the fact that it seems (and likely is) correct, I think it might actually be quite difficult to formulate a rigorous statement. As well as that, there seems to be a lot left to the reader. Why exactly does [K, M] = 0 imply [K^n, M^m] hold for positive powers n, m? Why do linear combinations of any powers also commute? Why do finite linear combinations extend straightforwardly to infinite linear combinations? These should all be made explicit. **Section 4**, which generalizes Sylvester flows, is difficult to parse and disentangle from previous work. - I'm not sure of the term 'generalized Sylvester flows' for this transformation: isn't a main selling point of the original Sylvester flow that the determinant of the Jacobian can be computed in O(M) instead of O(D) using Sylvester's identity, where generally M < D. You don't even need Sylvester's identity to derive eq. (9) since det dz / dx = det [I + W^-1 J_f(Wx) W] = det W det [I + W^-1 J_f(Wx) W] det W^-1 (since W invertible) = det [I + J_f(Wx) W W^-1] = det [I + J_f(Wx)] Unless I'm mistaken, where is Sylvester's identity being exploited? If it's not being used, why the Sylvester name? - Theorem 1: Following Papamakarios 2019 'Neural Density Estimation and Likelihood-free Inference' pg. 44-45, you have invertibility more succinctly by noticing that the Jacobian determinant det dz/dx = det(I + J_f) is the det of a triangular matrix and thus given by the product of diagonal elements, which are all positive by your assumption, so the inverse function theorem guarantees invertibility everywhere. This thesis is also worth a reference for the paragraph after the proof in that section, which states "Furthermore, if M = D, Q is taken to be a reverse-permutation matrix and R is strictly upper triangular, g becomes a MADE (Germain et al., 2015) with one hidden layer of D units. In that case, the Sylvester flow becomes a special case of the Inverse Autoregressive Flow (Kingma et al., 2016), where the scaling factor is 1 and the shifting factor is g(u).", and begins to draw the connection between Sylvester flows and autoregressive transformations that you have fleshed out more fully. - Sections 4.1 & 4.2: A lot starts to happen here ((i) the convolutional exponential, (ii) generalized Sylvester flows, and (iii) parameterization of orthogonal transformations with Householder reflections all coming together) and things become quite dense, without much description or analysis of e.g. the computational cost involved in any these operations. I also don't understand how this fixed-point iteration for computing an autoregressive inverse is compatible with 217-218 '...and since emerging convolutions are autoregressive, their inverse requires the solution to an iterative problem.' -- don't generalized Sylvester flows also require this? **Section 6.1**: - Following on from the previous point concerning the coming together of many different moving parts to form convolutional Sylvester flows, I think it would be beneficial to provide evaluation beyond just ELBO and NLL. Figure 3 in Lu & Huang 2020 provides an insightful look at wall-clock times for the various approaches (1x1, emerging, periodic, Woodbury). I'd like to see a similar comparison here given the relatively similar performance of the various methods.

Correctness: There doesn't seem to be anything obviously incorrect, although as mentioned, I'd like a more formal derivation of equivariance preservation under exponentiation, as well as a more careful presentation of the autoregressive residual transformation presented in Section 4. The empirical evaluation seems correct, and demonstrates the performance of the proposed transformations on both images and graphs.

Clarity: Generally I think the paper is well written, but I find sections 3.2 -- 4.2 somewhat opaque and difficult to penetrate - there's a lot going on, and I think it could be presented more clearly and carefully.

Relation to Prior Work: I think the contribution of the implicit and iterative exponential computation is well described and distinguished from previous work. I am less sure of how to disentangle the proposed generalized Sylvester flow from the previous work, and also confused as to whether this transformation even needs to be cast as a 'Sylvester' flow at all.

Reproducibility: Yes

Additional Feedback: 53: I think it's more correct to say the matrix exponential is well-defined because the series always converges, not vice versa. 58: Bold x. 63: 'where -> vectorizes x' I know you mean flatten an image x in R^(h x w x c) to a vector element in R^hwc, but 43 defines x as a vector element of R^d, so it's not clear what is meant here. 72: The first clause of this sentence is confusing, maybe change to 'As the main example we take the exponential of a convolutional layer, ...'? 81-82: This might be my own issue, but it wasn't immediately clear why this was true, and required some poring over of the figures. It might be worthwhile having a short note in the appendix analogously to how the trace of the graph convolutional exponential is detailed in 3.1 and the appendix. There also doesn't seem to be a discussion of kernels with spatial dimensions which are not odd? 128: Norm of M or || M ||, not norm of || M ||. 158-159: 'determinant of form z': z is a vector, you mean the determinant of the Jacobian of this map. 164: saying W is invertible after an equation where both W and its inverse appear is redundant. This should probably read 'Let W be invertible. Then the transformation is given by...'. ------------------------------------------------------- POST-REBUTTAL UPDATE ------------------------------------------------------- I'd like to thank the authors for their response. I've raised my score 5 -> 6, and would lean toward acceptance.


Review 2

Summary and Contributions: This work introduces a new method for forming bijective transforms from (not necessarily bijective) linear operations. The authors show that the matrix exponential of any linear operation is bijective, and we can calculate its forward and inverse operators using an iterative procedure up to a given degree of numerical precision. This opens up new possibilities for devising normalizing flows, and the authors present novel flows based on the idea for images and graphs. The authors demonstrate their new flows for density estimation on image and graph datasets, producing state-of-the-art results.

Strengths: The idea is simple but novel, and is well-explained. The method is technically sound, easy to implement, and supported by non-trivial experiments. As modeling of distributions on graphs is a hot topic right now, I believe this work is of great relevance and interest to the NeurIPS community.

Weaknesses: I can think of only a few weaknesses of this work. What can we say about the increase in computation due to the iterative procedure for the forward and inverse operator, as well as the need to apply spectral normalization? Does this make the comparison to other methods unfair? E.g. Graph NF to Graph NF + graph convolution exponential? Is spectral normalization really required or could you use weight normalization instead?

Correctness: Yes.

Clarity: Yes, the paper is well written and easy to understand. I think it could be improved by giving a brief discussion of spectral normalization in the supplementary materials so the work is self-contained (not sure how many readers will already be familiar with this method).

Relation to Prior Work: Yes.

Reproducibility: Yes

Additional Feedback: A question I have is, is the initialization of the three novel flows presented in the paper important and how are they initialized? There are quite a few typos in the supplementary materials: "l1" => "l_1", "sylvester" => "Sylvester", "an important difference that" => "an important difference is that", "a single after" => "a single time after", "the convolution exponentional" => "the convolution exponential", "Interestinly" => "Interestingly", "orginal" => "original", "distribtuions" => "distributions".


Review 3

Summary and Contributions: Normalizing flows allow density estimation via maximum-likelihood training and fast sample generation. The core of these approaches are flexible invertible mappings, which are usually implemented by invertible neural networks. A significant research body has focused on designing efficient and flexible neural network based flow models. This work contributes two new building blocks to this toolkit: 1) a linear flow via exponential mappings to the toolkit of flow building blocks and 2) a generalized Sylvester flow based on the exponential mapping and an auto-regressive function. Most notably, the proposed exponential mapping allows to turn any linear mapping into an invertible mapping, which is shown for convolutions and graph convolutions. On the experimental side, the work demonstrates improved performance when using their building block.

Strengths: The work introduces a new building block for normalizing flows, an area which received significant attention recently. Thus, the presented linear flow based on the exponential mapping will most likely be of interest for many deep learning practitioners as a drop-in replacement e.g. for 1x1 convolutions (from GLOW, Kingma et al. 2019). The main strengths are: • Theoretically justified approach to design invertible linear mappings (invertible for all matrices M, including convolutions, easy to invert, simple log-determinant computation) • Generalization of Sylvester Normalizing Flows (SNFs), including a proof that it generalizes previous SNFs, and convergence analysis of the fixed point iteration scheme for inversion. • Demonstration that this new building block improves performance over other linear flows and that the extended SNFs is practically more flexible.

Weaknesses: While being a solid contribution, the work has a few weaknesses: • I have some doubts about the Lipschitz analysis: ------ Section 4.1: In Appendix Line 49 it is stated that f_AR needs to be Lipschitz. However, I doubt that f_AR from line 191 (main body) is always Lipschitz: the product u*s_1(u) introduces a term u*ds_1/du when differentiating w.r.t. u. If u --> infinity, then ds_1/du needs to go to zero, otherwise the term u*ds_1/du will tend to infinity, which makes the mapping f_AR not Lipschitz. I expect one needs some additional constraints to ensure ds_1/du --> 0. Please comment on this on how exactly the Lipschitzness of f_AR is required. ------ Related to above: in line 67 (appendix) it is claimed that the Lipschitz constant L of f_AR can be easily controlled. However, the mapping f_AR seems rather complex and is not only a composition of Lipschitz continuous components, since it involves multiplicative interactions. Please comment. ------ The \gamma in line 191 (main paper) and \gamma in line 53 (appendix) are not necessarily the same, right? The Lipschitz constant \gamma can be different to the factor \gamma. • There is no runtime comparison, some complexity analysis is missing: ------Please provide some more details on the additional computational complexity of the proposed conv-exp flows. In particular, how are the wall-clock runtimes in the experiments from section 6.1? I´d expect that the power iteration and the spectral normalization introduces some overhead over competing approaches. ------Line 130: "graph convolution exponential converges quickly" - make this more precise by quantitative results • Some discussion of related work is missing: ------In Li et al. (2019, NeurIPS) "Preventing gradient attenuation in lipschitz constrained convolutional networks" orthogonal convolution layers are studied which could also be used as linear flows. Their approach is also iterative (Björck Orthogonalization). Please discuss connections. ------ In Behrmann et al. (2019, ICML) "Invertible Residual Networks" the matrix-logarithm was used implicitly for log-determinant estimation of the Jacobian, which is conceptually very similar. • There is no discussion of the expressivity of the proposed approach: ------ Can the exponential mapping with an arbitrary matrix M represent all invertible matrices? If not, which matrices cannot be represented? ------ In Appendix Line 91 it is stated that conv-exp mixing is used with an additional invertible 1x1 convolution to increase flexibility (otherwise conv-exp to close to identity). This seems to be an important downside of conv-exp and I wish the authors would have studied the expressivity of conv-exp and potential downsides more explicitly in the main body of the paper. To summarize, there appear to be some flaws, which I hope can be addressed in the rebuttal.

Correctness: The proofs and proposed method appear to be correct. A few doubts appeared with respect to the Lipschitz properties of the autoregressive mapping within the Generalized Sylvester Flow (see details in "weaknesses). The empirical methodology appears to be solid.

Clarity: The paper is overall well written and splits main concepts/ technical details well between main body/ appendix. Furthermore, some visualizations help to understand the proposed approach. In order to further ease the understanding, the paper could have included an example of how the linear mapping changes under the exponential mapping. For example it would be interesting to see how a simple filter like an edge detector gets transformed during the power series.

Relation to Prior Work: The discussion of related work is appropriate and highlights major differences. However, some related work discussion is missing, see "weaknesses".

Reproducibility: Yes

Additional Feedback: Below are some further questions and minor issues: • Why are the experiments from Appendix D not even referenced in the main body? • Line 130: "graph convolution exponential converges quickly" - make this more precise by quantitative results • Line 56: include reference to Theorem in Linear Algebra textbook to aid readers that want to refresh their background on matrix exponentials • Section 2.2: add reference to Sedghi et al. (ICLR 2019): "The singular values of convolutional layers" for more background on Convolutional-matrices • Add details on why Spectral norm (Miyato et al. 2018) only produces a lower bound. It is straightforward to extend this approach to the full matrix M, which will yield approximates ||M||_2, see Gouk et al. (2018) "Regularisation of Neural Networks by Enforcing Lipschitz..." • Line 126: add exact reference "Appendix B" • Line 139: not clear here what "capsules" are • Line 158: "determinant of form..." - Jacobian determinant of mapping z=.... ? • Line 162: "improvement of Sylvester flows" - in which way does this improve SNFs? Maybe call it "extension"? • Line 56 (appendix): \ell_1 • Line 221: not sure why Neural ODEs (Chen et al. 2018) need to be cited when speaking about linear ODEs. • Line 131 (appendix): typo "interestingly" • Line 56 (appendix): "prove" --> proof ------------------------------ Post-Rebuttal update: ------------------------------ Thank you for the detailed discussion of our raised points. I think the revised manuscript will benefit from these additional explanations and corrections. However, these changes are somewhat substantial (changes in proofs and derivations, additional experiments), which is why I only increase my score to 6.


Review 4

Summary and Contributions: The paper proposes a new way to construct invertible transformations for flow models by taking an exponential of a linear transformation such as a convolution. The proposed way is attractive as a building block of flow models as it allows for a compute and memory efficient calculation of the log determinant of the transformation as well as the transformation itself. The flexibility and benefits of the convolution exponential are demonstrated empirically on a standard image dataset and toy graph datasets. Finally, the paper also proposes Generalized Sylvester Flows, whose performance is also demonstrated empirically.

Strengths: The work presents several novel ideas that would be of interest to the scientific community - in particular the convolution exponential and the generalization of the Sylvester flows. The claims are supported through theoretical grounding and empirical results and analyses; specifically, results in Table 1 comparing the various mixing operations would be interesting for the community.

Weaknesses: While the work proposes a wealth of ideas (convolution exponential, graph convolution exponential, generalization of sylvester flows and convolutional sylvester flows), it lacks somewhat in the depth of presentation of these ideas. The empirical evaluation of the methods is limited (e.g. only a single small-scale dataset used for image experiments). In fact the work could be easier to present if it were split into the two contributions mentioned in the title - the convolution exponential and the generalized Sylvester flows; that should also provide more experimental results and details. For example, it would be interesting to understand how competitive the convolution exponential is on other image datasets in terms of NLL, as well as the quality of the generated images - does the use of a convolution exponential in place of other mixing operations provide an inductive bias useful for generation?

Correctness: As far as I can tell, yes.

Clarity: The paper proposes several sufficiently independent ideas: the convolution exponential, and the generalized Sylvester flows; and the graph convolution exponential, which could be viewed as an independent idea. The authors did an impressive job weaving these three ideas together in a single story, but the resulting paper is still dense and broad due to the amount of material it tries to cover in the limited space available. The paper is well-written overall, but it glances over some prior knowledge here and there. For example, * Section 3.2 talks about feature fields and capsules without explaining what these are and how they relate to each other. * Overall, section 3.2 could use more background information, as well as motivation for its investigations. The authors may consider moving the details of this section into the appendix and instead focusing on the implications of the equivariance of the convolution exponential. Also the Supplementary information contains a few typos (e.g. “Interestinly”) and experiments (Section D) that do not seem to be referenced anywhere in the main text.

Relation to Prior Work: Yes. A few minor points: * “Unlike prior work, we observe that the 36 exponential can be computed implicitly” should include a citation. * “The matrix exponential has been largely used in the field of ODEs”, could also use some citations. * The relationship to i-ResNets could be discussed more. i-ResNets and the convolution exponential are somewhat dual. Both use spectral normalization and interactive procedures, but in different ways and for computing the different parts of the flow.

Reproducibility: Yes

Additional Feedback: ------------------------------------------------------- POST-REBUTTAL UPDATE ------------------------------------------------------- I'd like to thank the authors for their response and the changes they are making to the manuscript. However, I do not believe that the changes warrant increased the score at this time.

[Author Response · NeurIPS 2020]

We sincerely thank all reviewers for their detailed feedback and the positive comments indicating that we present a novel idea (R2, R4) of great relevance for the community (R2) with an empirical evaluation that demonstrates the performance of the method on both images and graphs (R1). We will clarify the issues raised by the reviewers in our response below.

*Firstly R1, R2, R3 are interesteed in runtime of the convexp.* We agree that this can be valueable and we will include a runtime comparison experiment with a table of the different linear flows in our updated manuscript. In short, the increase in computation is straightforward: we typically need 6 convolutional calls (Fig. 3), but this is somewhat balanced by a cheap determinant. Our tests show that the runtime of the convexp-flow utilizes 10.9% more computation time during training than a flow using 1x1 convolutions.

*R1, R4 note that the section on equivariance is lacking clarity, because the notation in this section is somewhat different.* We agree, and we will clean up this notation and remove terms that are not necessary to understand the section such as capsules and feature field. The main idea of this section is to prove that the exponential convolution preserves the equivariant properties of its underlying convolutions. *R1* asks why $[K, M] = 0$ implies $[K^n, M^m]$. The result can be derived as follows: $[A, BC] = ABC - BCA = ABC - BAC + BAC - BCA = [A, B]C + B[A, C]$. Higher powers follow from induction. Our claim $[K, \exp M] = 0$ can also be derived straightforwardly: Define $\exp_n$ as the exponential taking only the first $n$ terms of the series. Since $0 = \lim_{n\to\infty}[\exp_n M, K] = [\lim_{n\to\infty} \exp_n M, K] = [\exp M, K]$ by continuity of $[..., ...]$. These intermediate steps will also be clarified in the manuscript. We will also make a note of the connection with Lie algebras and Lie groups. By request of R4 we will also focus more on the intuitive implications.

*R1 is asking whether the determinant of Sylvester flows could be derived without Sylvester's identity, and asks whether Sylvester flows always have dimensionality reduction.* Note that the original Sylvester flows come in three flavours: Although O-SNFs reduce dimensions, H-SNFs and T-SNFs do not. Further, there are indeed multiple derivations that give the determinant in Eq. 9. Notice that the derivation suggested by R1 is very similar to our "Remark II, App. A" which gives an alternate proof for invertibility (which also applies to H/T-SNFs). Due to the similarity between our extension and H/T-SNFs, we decided to call them generalized Sylvester Flows. To clarify our manuscript, we will include the derivation suggested by R1 and write Thm. 1 more succinctly using (Papamakarios, 2019).

*R3 questions whether $f_{\mathrm{AR}}$ is indeed L-Lipschitz.* Firstly, note that this is for an arbitrarily high constant $L$ (which makes it a rather weak constraint). The reason why we require this, is so that $\gamma^t \cdot L \to 0$ eventually, and the FPI converges. R3 is correct that in theory on $\mathbb{R}$ the function may not be Lipschitz, caused by the product $s_1(u) \cdot u$. However, computer signals generally have bounded domains, and $f_{\mathrm{AR}}$ is already $L$-Lipschitz on these bounded domains. The theoretical issue can be solved by altering the transformation slightly: we can clip/threshold the variable $u$ which is multiplied with $s_1$, but only for *very high* magnitudes. Since $s_1, s_2$ are bounded by a $\tanh$ and $u$ is now also bounded, the function $f_{\mathrm{AR}}$ is now Lipschitz even for $\mathbb{R}$. Note that this modifications leaves all experimental results valid, as these large values were practically never reached. Although it may be expensive to compute $L$ explicitly, it can be steered by limiting the Lipschitz continuity of $s_1, s_2, t_1, t_2$. Further, $\gamma$ should be seen as an upper bound on the magnitude of the diagonal of $\mathbf{J}_{f_{\mathrm{AR}}}$. Since $s_1, s_2$ are bounded $(-1, 1)$ and are strictly triangular functions, the $\gamma$ in the main paper is an upper bound of this diagonal. This discussion will be added to the inverse analysis in the appendix.

*R3 asks about the expressitivity of the exponential.* The output of the exponential $(\exp M)$ can be any matrix that is the solution to the linear ODE of the form $\dot{x} = \mathbf{M}x$ from $t = 0$ to $t = 1$. It cannot model all invertible matrices, for instance, matrices with negative determinants cannot be modelled. Further, the spectral normalization constraints the matrix $\mathbf{M}$ in the possible linear ODEs. This will be included in the manuscript.

*Other comments/questions | R2 asks about replacing spectral norm with weight norm.* Although weight norm could also improve the series convergence, spectral norms give theoretically guaranteed convergence behaviour as shown in Fig. 3. | *R3 asks for a discussion on connections related work.* We will explain the connection to computing the logarithm of a matrix (which in contrast with the exp cannot always be computed in a stable fashion and converges slowly) from (Behrmann et al.) and the orthogalization procedure in (Li et al. 2019). | *R1 notes that Emerging convs and Sylvester flows both need to be solved iteratively.* Correct, but in L217-218 different linear flows are compared. This iterative inverse of emerging convs would make them impractical to use as basis change inside Sylvester flows as both inverse and forward of the linear flow are required during optimization. | *R2 asks how methods were compared.* In image experiments, we adjusted the size of the coupling layers to ensure a roughly equal parameter budget. For graph experiments, the coupling layers were kept the same as the convexp only added a negligible number of parameters $(< 0.01\%)$. | *R3 is concerned that convexp are combined with 1x1 convs.* Note that 1x1 convs can be made cheap, especially when modelled using Householder transformations (as we proposed for Sylvester Flows). We will include a discussion of the tendency of convexp to remain close to the identity in the main text and connect it to the limited induced matrix norm. | *R3 suggests to add details on spectral normalization.* We will extend the discussion of spectral norms using (Gouk et al.) | *R1, R2, R3, R4:* Beside the already mentioned changes we also fixed all minor issues that reviewers spotted in the paper (e.g. typos, tips to better phrase some sentences and references to the appendix.)

[Meta-Review · NeurIPS 2020]

Good paper that introduces an alternative way of parameterizing an invertible linear transformation using the exponential map of an unconstrained matrix. Even though this parameterization has been considered before, the idea of not instantiating the linear transformation but instead computing directly its action on the input vector by Taylor-approximating the exponential map is neat and computationally efficient. The reviewers raised a few concerns, which the rebuttal addresses. I would encourage the authors to take to heart the reviewer's comments and suggestions for improvement when preparing the camera-ready version.